# Resolution of Halogenated Mandelic Acids through Enantiospecific Co-Crystallization with Levetiracetam

**DOI:** 10.3390/molecules26185536

**Published:** 2021-09-12

**Authors:** Jie Wang, Yangfeng Peng

**Affiliations:** School of Chemical Engineering, East China University of Science and Technology, Shanghai 200237, China; Y30180872@mail.ecust.edu.cn

**Keywords:** co-crystal, resolution, halogenated mandelic acid, 3-chloromandelic acid, levetiracetam

## Abstract

The resolution of halogenated mandelic acids using levetiracetam (LEV) as a resolving agent via forming enantiospecific co-crystal was presented. Five halogenated mandelic acids, 2-chloromandelic acid (2-ClMA), 3-chloromandelic acid (3-ClMA), 4-chloromandelic acid (4-ClMA), 4-bromomandelic acid (4-BrMA), and 4-fluoromandelic acid (4-FMA), were selected as racemic compounds. The effects of the equilibrium time, molar ratio of the resolving agent to racemate, amount of solvent, and crystallization temperature on resolution performance were investigated. Under the optimal conditions, the resolution efficiency reached up to 94% and the enantiomeric excess (%e.e.) of (*R*)-3-chloromandelic acid was 63%e.e. All five halogenated mandelic acids of interest in this study can be successfully separated by LEV via forming enantiospecific co-crystal, but the resolution performance is significantly different. The results showed that LEV selectively co-crystallized with *S* enantiomers of 2-ClMA, 3-ClMA, 4-ClMA, and 4-BrMA, while it co-crystallized with *R* enantiomers of 4-FMA. This indicates that the position and type of substituents of racemic compounds not only affect the co-crystal configuration, but also greatly affect the efficiency of co-crystal resolution.

## 1. Introduction

Chiral compounds are an important class of organic compounds, whose single enantiomers are widely used in pharmaceutical, agricultural, and food industries, and also have application prospects in electronic devices and liquid crystals [1,2,3,4]. Single enantiomer can be obtained by the synthesis method or chiral resolution. Synthesis routes include chiral pool synthesis (using enantiopure starting materials) and asymmetric synthesis (using chiral auxiliary or catalyst) [5,6,7,8,9]. Although these two synthetic methods have obvious advantages in obtaining a single enantiomer, the limited number of sources of enantiopure starting materials and high cost are not conducive to a wide application of chiral pool synthesis, while asymmetric synthesis requires a long development time and high cost [10,11,12,13,14,15]. Chiral resolution is a method of selectively separating one enantiomer from the racemate to obtain the target enantiomer by various techniques, such as inclusion resolution [16], liquid–liquid extraction [17], electrophoresis [18], biological resolution [19], and membrane technology [20]. However, these resolution methods have some disadvantages, such as high cost and small scale, which limits their wide application [21,22].

Up to now, formation of diastereomeric salt [23,24,25,26,27] and chiral chromatography [28,29,30,31] are two commonly applied chiral resolution methods in industry. For those compounds of interest that are easy to form salts, the formation of diastereomeric salt is the most suitable method, as a chiral resolving agent can form a pair of diastereomer salts with two enantiomers that show difference in their physical properties, promoting the success of the resolution [32]. Chiral chromatography, though expensive, seems to be the only industrially viable technology for compounds that cannot form or are not easy to form salts. Co-crystal resolution emerges as a novel chiral resolution method, which depends on the hydrogen bonding between two chiral compounds, and can be used as an alternative resolution method for those compounds that cannot form or are not easy to form salts. Different from the formation of diastereomeric salts, co-crystal resolution generally shows enantiospecific behavior, with one of the two enantiomers selectively forming co-crystal with chiral resolving agents [33], which may lead to a high yield in a single crystallization step. This method has great potential to be applied to a variety of racemic compounds that cannot be effectively resolved by diastereomeric salt formation.

Pharmaceutical co-crystals have attracted more and more attention as they can improve the physical and chemical properties of pharmaceuticals, including solubility and dissolution rate, thermal stability, bioavailability, and hygroscopicity [34,35,36]. Separation of racemates by forming enantioselective co-crystal has also been studied recently in the pharmaceutical industries. Springuel et al. [37] successfully resolved (*R*, *S*)-2-(2-oxopyrrolidine-1-yl) butyramide by enantiospecific co-crystallization of (*S*)-mandelic acid ((*S*)-MA) and (*S*)-2-(2-oxopyrrolidine-1-yl) butyramide (the trade name is levetiracetam, or LEV for short), which showed a high efficiency, as 70% of the *S*-enantiomer were separated from the racemic mixture in a single co-crystallization step. Sánchez et al. [38] obtained a diastereoisomeric co-crystal pair of *L*-malic acid (*L*-MA) and praziquantel (PZQ) by a liquid assisted grinding method. The crystal structure showed that (*R*)-PZQ-L-MA had stronger intermolecular interaction than (*S*)-PZQ-L-MA, and the maximum %e.e. of (*R*)-PZQ after co-crystal resolution is as high as 99.3%e.e. However, there have rarely been studies reported on chiral resolution by co-crystallzation. Therefore, more research on co-crystal resolution is required to provide guidance for separating compounds that cannot form or are not easy to form salts.

LEV is a widely used antiepileptic drug [37,39] with a relatively low processing cost compared with some traditional resolving agents. In particular, LEV is considered as a potential good co-crystal resolving agent as it has two amido groups, which can easily form co-crystals with a co-crystal precursor. Springuel et al. [37] indicated that (*S*)-MA can be used to resolve (*R*, *S*)-2-(2-oxopyrrolidine-1-yl) butyramide, whereas LEV can also resolve racemic mandelic acid. Because mandelic acid can be easily substituted by benzene ring to obtain many kinds of its derivatives, studying the co-crystal resolution of mandelic acid derivatives with LEV is beneficial to find out and develop the law of co-crystal resolution. Therefore, LEV was selected as a resolving agent in enantiospecific co-crystallization to resolve racemic 2-chloromandelic acid (2-ClMA), 3-chloromandelic acid (3-ClMA), 4-chloromandelic acid (4-ClMA), 4-bromomandelic acid (4-BrMA), and 4-fluoromandelic acid (4-FMA). Optically active halogenated mandelic acid is an important pharmaceutical intermediate, chiral resolving agent, and chiral ligand. Meanwhile, halogenated mandelic acids have hydroxyl and carboxyl groups, which can easily form salt or co-crystal with other compounds.

According to our previous work [40], or see Appendix A, as illustrated in Figure 1, (*S*)-2-chloromandelic acid ((*S*)-2-ClMA), (*S*)-3-chloromandelic acid ((*S*)-3-ClMA), (*S*)-4-chloromandelic acid ((*S*)-4-ClMA), and (*S*)-4-bromomandelic acid ((*S*)-4-BrMA) were able to co-crystallize with LEV, while no co-crystallization occurred for (*R*)-2-chloromandelic acid ((*R*)-2-ClMA), (*R*)-3-chloromandelic acid ((*R*)-3-ClMA), (*R*)-4-chloromandelic acid ((*R*)-4-ClMA), and (*R*)-4-bromomandelic acid ((*R*)-4-BrMA) with LEV. However, 4-FMA showed different characteristics from 2-ClMA, 3-ClMA, 4-ClMA, and 4-BrMA. (*R*)-4-Fluoromandelic acid ((*R*)-4-FMA) formed a co-crystal with LEV, while no co-crystal formation occurred when combining (*S*)-4-fluoromandelic acid ((*S*)-4-FMA) with LEV. Five co-crystals were characterized by X-ray powder diffraction (XRPD), differential scanning calorimetry (DSC), infrared spectroscopy (IR), solid state nuclear magnetic resonance (^13^C NMR), and elemental analysis (EA) [40].

In this study, the effects of resolution conditions on the resolution effect were investigated in detail, including equilibrium time, molar ratio of the resolving agent to racemic compound, solvent amount, and crystallization temperature. Furthermore, the influence of position and type of substituents of racemic compounds on chiral resolution [41,42] was also studied. This study not only successfully resolved several halogenated mandelic acids, some in high yield, but also provided reference and fundamental support for co-crystal resolution of racemic compounds that could not form or are not easy to form salts.

## 2. Results and Discussion

Solvent is an important factor affecting co-crystal formation. Co-crystal is formed by two or more different substances depending on non-covalent bonds such as hydrogen bonding, π–π stacking, and Van der Waals force [43]. There is no proton transfer during co-crystal formation, which is different from salt formation, so the interaction between different substances forming co-crystal is weaker than that between different substances forming salt [44,45]. Because hydrogen bonding is one of the main forces to form co-crystal, co-crystal can not be formed in solvents with strong hydrogen bonding between molecules. Springuel et al. [37,46] reported that mandelic acid and LEV formed co-crystal in acetonitrile. Our preliminary study also showed that, under the same experimental conditions, levetiracetam and 3-ClMA could not form co-crystal precipitated solid in methanol, ethanol, and isopropanol, and only a small amount of solid could be obtained when ethyl acetate was used as solvent for co-crystal resolution. In this work, combined with the research of Springuel et al. [37,46], the resolution of mandelic acid derivatives by LEV used acetonitrile as solvent.

### 2.1. Effect of Equilibrium Time on Co-Crystal Resolution

As shown in Figure 2, the effect of equilibrium time on co-crystal resolution was investigated in this study, taking the resolution of 3-ClMA by LEV as an example. In general, with the increase in equilibrium time, the optical purity of (*S*)-3-ClMA in the solid phase, resolution efficiency, and %e.e. for (*R*)-3-ClMA in the liquid phase gradually increased, but some errors were caused by the vacuum filtration process. When the equilibrium time was one day, the resolution efficiency reached 47%, which is already a good result. An equilibrium time of 12 days was chosen to obtain the best experimental results. At this time, the optical purity of (*S*)-3-ClMA in the solid phase was up to 81%, resolution efficiency reached 69%, and %e.e. for (*R*)-3-ClMA in the liquid phase was 58%. No significant benefits were observed prolongating equilibrium time beyond 12 days.

### 2.2. Effect of Molar Ratio on Co-Crystal Resolution

In the process of co-crystal resolution, the molar ratio of the resolving agent to racemate has a great influence on the resolution results. It can be seen from Figure 3 that the resolution efficiency and %e.e. in the liquid phase increased first and then decreased with the increase in molar ratio. When the ratio of the resolving agent LEV is low, the hydrogen bonds and other interactions between 3-ClMA molecules may lead to low resolution efficiency; when the ratio of LEV exceeds a certain point, LEV will crystallize out of the solution because its solubility is much lower than that of 3-ClMA, and the concentration of LEV in the solution will be greatly reduced, while the content of LEV in the solid phase will be relatively high. The experimental results show that the resolution efficiency reached the highest when the molar ratio was 55:45, thus the molar ratio of 55:45 was chosen in the following investigation.

### 2.3. Effect of Amount of Solvent on Co-Crystal Resolution

The amount of solvent is also a factor affecting co-crystal resolution, and its influence on resolution effect was investigated. When the molar fraction of acetonitrile was 97 mol% and 93 mol%, a small quantity of 3-ClMA that was slightly undissolved (negligible) in acetonitrile was filtered by the needle filter, which had a certain influence on the %e.e. in the liquid phase. As shown in Figure 4, with the increase in the mole fraction of acetonitrile, and the resolution efficiency and %e.e. in the liquid phase first increased and then decreased. The experimental results show that the resolution efficiency was obviously the highest when the mole fraction of acetonitrile was 93 mol%, and the corresponding optical purity was also relatively high. Therefore, 93 mol% was chosen as the mole fraction of acetonitrile in this study.

### 2.4. Effect of Crystallization Temperature on Co-Crystal Resolution

Besides the equilibrium time, molar ratio, and amount of solvent, crystallization temperature is another critical factor affecting the resolution process. After determining the above optimum conditions, the crystallization temperature was examined to further improve the resolution efficiency. As shown in Figure 5, with the decrease in crystallization temperature, the optical purity of (*S*)-3-ClMA in the solid phase decreased; on the contrary, the resolution efficiency increased. However, when the crystallization temperature dropped to −22 °C, the optical purity increased slightly and the resolution efficiency decreased. Therefore, the crystallization temperature of −18 °C was chosen as the most suitable crystallization temperature, at which the resolution efficiency was as high as 94%, the optical purity reached 83%, and %e.e. for (*R*)-3-ClMA in the liquid phase was 63%.

### 2.5. Resolution of Other Halogenated Mandelic Acids by LEV

Considering the effective resolution results of 3-ClMA by forming enantiospecific co-crystal using LEV as a resolution agent, 2-ClMA, 4-ClMA, 4-BrMA, and 4-FMA were selected as other halogenated mandelic acid materials for further investigation of the effect of substituent position and type of racemic compounds on co-crystal resolution. The formation of co-crystal depends on the molecular interaction between halogenated mandelic acids and LEV molecules. Owing to various positions and types of substituents of halogenated mandelic acids, the racemic compounds may show different enantiomeric selectivity and resolution efficiency towards LEV. As shown in Table 1, the co-crystals of *S* enantiomer of halogenated mandelic acids and LEV were obtained when LEV was used as the resolving agent to resolve 2-ClMA, 4-ClMA, and 4-BrMA, which is the same as the result of LEV resolving 3-ClMA. However, the co-crystal of (*R*)-4-FMA and LEV was obtained in the resolution of 4-FMA. These show that LEV has better recognition ability for *S* enantiomers in 2-ClMA, 3-ClMA, 4-ClMA, and 4-BrMA than for *R* enantiomers, but preferably recognizes (*R*)-4-FMA other than (*S*)-4-FMA. Therefore, the substituent type of racemic compound can affect the stereo configuration of products separated by forming co-crystal. Table 1 shows the results without optimization of co-crystal resolution conditions. The optical purities of the products obtained using LEV as the resolving agent by co-crystal formation of four halogenated mandelic acids are above 70%, which indicates that LEV can effectively resolve the four halogenated mandelic acids. The resolution efficiency and %e.e. for 2-ClMA with LEV as the chiral resolving agent are much higher than for 4-ClMA, 4-BrMA, and 4-FMA, which indicates that the position and type of substituents of racemic compounds will affect the efficiency of co-crystal resolution. In addition, without optimization of the resolution conditions of 2-ClMA and 4-ClMA, LEV has the highest resolution efficiency on 3-ClMA, followed by 2-ClMA, and 4-ClMA has the lowest resolution efficiency. Under the experimental scope of this study, the resolution efficiency is in the order of 2-ClMA > 3-ClMA > 4-ClMA. In other words, 3-ClMA is the one to easily be resolved by LEV, while 4-ClMA is the hardest one to resolve.

## 3. Materials and Methods

### 3.1. Materials

LEV with an optical purity of 99% was supplied by Hangzhou Zequan Biology Science and Technology Co., Ltd. (Zhejiang, China). 2-ClMA and 4-ClMA with 98% purity were purchased from Shanghai Dongyue Drug Co., Ltd. (Shanghai, China). 3-ClMA, 4-BrMA, and 4-FMA with 98% purity were purchased from Shanghai Titan Scientific Co., Ltd. (Shanghai, China). L-leucine and ZnSO_4_·7H_2_O with 99% purity were purchased from Shanghai Titan Scientific Co., Ltd. (Shanghai, China). Acetonitrile was analytically pure grade with 99% purity from Shanghai Titan Scientific Co., Ltd. (Shanghai, China). Methanol was chromatographically pure grade with 99.9% purity from Shanghai Titan Scientific Co., Ltd. (Shanghai, China). Water was purchased from Hangzhou Wahaha Co., Ltd. (Zhejiang, China). All chemicals were used without any further purification.

### 3.2. Sample Analysis

The optical purity of halomandelic acid enantiomers in the solid/liquid phase obtained by co-crystal resolution was determined by HPLC analysis. The HPLC analysis was performed on a ST150000 high-performance liquid chromatograph (Hegong Scientific Instruments, Shanghai, China) equipped with a UV (STI UV5000) detector and C18 column (length of 250 mm, internal diameter of 4.6 mm, particle size of 5 μm) purchased from Tianjin Beisile Chromatography Technology Development Center, China. The injection volume was 10 μL. The mobile phase consisted of methanol and aqueous solution (1:9, *v/v*) containing 8 mmol·L^−1^ L-leucine and 4 mmol·L^−1^ Zn^2+^ with pH of 6.5, which was employed at a flow rate of 0.8 mL·min^−1^. The detection wavelength was set at 215 nm. The relevant HPLC diagrams for the separation and detection of five halogenated mandelic acids areshown in Figure 6.

The content of LEV of separated solid samples was determined based on HPLC analysis with an external standard method, and the correlation coefficient of the standard curve was 0.9994, which showed that the correlation was good.

The mass percentage content of LEV in co-crystal of LEV and halogenated mandelic acid (solid phase) obtained by resolution (LEV%), optical purity of *S* enantiomer or *R* enantiomer of halogenated mandelic acids in the solid phase (OS% and OR%), resolution efficiency (E%), and enantiomeric excess value (%e.e.) can be calculated by the following equations.
(1)$$\mathrm{LEV}\%=\frac{c_{2}}{c_{1}}\times 100\%$$
(2)$$\mathrm{OS}\%=\frac{SS}{SS+SR}\times 100\%$$
(3)$$\mathrm{OR}\%=\frac{SR}{SS+SR}\times 100\%$$
(4)$$E\%=\frac{m_{2}\times \left(1-LEV\%\right)\times OS\%}{m_{1}÷2}\times 100\%\;\;\mathrm{or}\;E\%=\frac{m_{2}\times \left(1-LEV\%\right)\times OR\%}{m_{1}÷2}\times 100\%$$
(5)$$e.e.\%=\frac{SR-SS}{SS+SR}\times 100\%\;\mathrm{or}\;e.e.\%=\frac{SS-SR}{SS+SR}\times 100\%$$
where $c_{1}$ (mg·L^−1^) represents the concentration of solid samples obtained by resolution to perform HPLC analysis. $c_{2}$ (mg·L^−1^) represents the LEV concentration obtained from the standard curve determined by HPLV analysis. SS (uV·S) and SR (uV·S) represent peak areas of *S* enantiomer and *R* enantiomer of halogenated mandelic acids in the solid phase obtained by resolution measured by HPLC, respectively. $m_{1}$ (g) represents the mass of raw material racemic halogenated mandelic acid. $m_{2}$ (g) represents the mass of the solid sample obtained by resolution.

### 3.3. The Procedure of LEV Resolving 3-ClMA

The typical resolution process is described as follows: 0.433 g (2.32 mmol) 3-ClMA was added into a 20 mL straight screw bottle containing 3.5 mL acetonitrile, and the suspension was heated to 75 °C to completely dissolve the solid under agitation for one and half an hours, followed by adding 0.483 g (2.84 mmol) LEV. The mixture was stirred at 75 °C and settled for 3 h. The obtained homogeneous solution was slowly cooled at room temperature and seeded with a co-crystal obtained by liquid-assisted grinding, and then allowed to stand at −18 °C for 12 days to achieve solid–liquid equilibrium. The precipitated crystalline co-crystal was collected by vacuum filtration and washed with acetonitrile. The solid was measured to be 0.490 g, and the optical purity of *S* enantiomer of 3-ClMA measured by HPLC reached 83%. The %e.e. was 63% for (*R*)-3-ClMA in the liquid phase, which was analyzed by HPLC.

### 3.4. The Procedure of LEV Resolving 2-ClMA

At ambient temperature and under agitation, 0.636 g (3.73 mmol) LEV was added to a solution containing 0.586 g (3.14 mmol) 2-ClMA completely dissolved in 2.0 mL acetonitrile. The mixture was heated to 60 °C under agitation for two hours to completely dissolve the solid. Then, the obtained homogeneous solution was slowly cooled at room temperature and seeded with a co-crystal obtained by liquid-assisted grinding, and stored at −15 °C for 14 days to achieve solid–liquid equilibrium. The precipitated crystalline co-crystal was collected by vacuum filtration and washed with acetonitrile. The dried solid was 0.760 g, and the optical purity of *S* enantiomer of 2-ClMA measured by HPLC was 78%. The %e.e. reached up to 76% for (*R*)-2-ClMA in the liquid phase, which was measured by HPLC.

### 3.5. The Procedure of LEV Resolving 4-ClMA

Here, 0.529 g (2.84 mmol) 4-ClMA was placed into a 20 mL straight screw bottle containing 2.2 mL acetonitrile, and the suspension was heated to 80 °C under agitation for one and half an hours, followed by removing a small amount of undissolved solid with a needle filter. Then, 0.395 g (2.32 mmol) LEV was added to the above solution and stirred at reflux temperature for 3 h to completely dissolve the solid. Then, the obtained solution was slowly cooled at room temperature and seeded with a co-crystal obtained by liquid-assisted grinding, and kept at −15 °C for 14 days. The precipitated crystalline co-crystal was collected by vacuum filtration and washed with acetonitrile. The solid was measured to be 0.190 g, and the optical purity of *S* enantiomer of 4-ClMA measured by HPLC was up to 88%. The %e.e. was 23% for (*R*)-4-ClMA in the liquid phase, which was analyzed by HPLC.

### 3.6. The Procedure of LEV Resolving 4-BrMA

Here, 0.726 g (3.14 mmol) 4-BrMA and 0.636 g (3.73 mmol) LEV were placed into a 20 mL straight screw bottle containing 2.0 mL acetonitrile, and the mixture was heated to 80 °C under agitation for one and half an hours, followed by filtration to remove a small amount of undissolved solid with a needle filter. Then, the filtered solution was slowly cooled at room temperature and seeded with a co-crystal obtained by liquid-assisted grinding, and placed at −15 °C for 14 days. The precipitated crystalline co-crystal was collected by vacuum filtration and washed with acetonitrile. The solid was measured to be 0.380 g, and the optical purity of *S* enantiomer of 4-BrMA measured by HPLC reached 71%. The %e.e. was 21% for (*R*)-4-BrMA in the liquid phase, which was analyzed by HPLC.

### 3.7. The Procedure of LEV Resolving 4-FMA

Here, 0.526 g (3.09 mmol) 4-FMA was placed into a 20 mL straight screw bottle containing 1.8 mL acetonitrile, and the suspension was heated to 80 °C under agitation for one and half an hours, followed by filtration to remove a small amount of undissolved solids with a needle filter. Then, 0.351 g (2.06 mmol) LEV was added to the above solution and stirred at reflux temperature for 3 h to completely dissolve the solid. Then, the obtained solution was slowly cooled at room temperature and seeded with a co-crystal obtained by liquid-assisted grinding, and placed at −15 °C for 14 days. The precipitated crystalline co-crystal was collected by vacuum filtration and washed with acetonitrile. The solid was measured to be 0.340 g, and the optical purity of *R* enantiomer of 4-FMA measured by HPLC was up to 90%. The %e.e. was 29% for (*S*)-4-FMA in the liquid phase, which was analyzed by HPLC.

## 4. Conclusions

In this study, a novel co-crystallization process for separating racemic 2-ClMA, 3-ClMA, 4-ClMA, 4-BrMA, and 4-FMA was proposed, namely co-crystal resolution using LEV as a resolving agent. Taking 3-ClMA as an example, the effects of the equilibrium time, molar ratio of the resolving agent to racemate, amount of solvent, and crystallization temperature on resolution performance were studied. The optimal resolution conditions were as follows: the equilibrium time was 12 days, the molar ratio of the resolving agent to racemate was 55:45, the amount of solvent was 93 mol%, and the crystallization temperature was −18 °C. Under these optimal conditions, the resolution efficiency reached 94% and %e.e. for (*R*)-3-ClMA in the liquid phase was 63%. This study also finds that 2-ClMA, 3-ClMA, 4-ClMA, 4-BrMA, and 4-FMA can be successfully resolved by LEV, but the resolution efficiency is obviously different. Without optimization of 2-ClMA and 4-ClMA, LEV shows the best resolution result on 3-ClMA, followed by 2-ClMA, and 4-ClMA has the lowest resolution efficiency. The resolution performance of 4-FMA is better than those of 4-ClMA and 4-BrMA. These results indicate that the resolution efficiency is highly related to the position and type of substituents of racemic compounds. In addition, LEV resolved 2-ClMA, 3-ClMA, 4-ClMA, and 4-BrMA via co-crystallization to obtain *S*-enantiomers, but resolved 4-FMA to obtain *R*-enantiomer, which indicates that the position and type of substituents of racemic compounds can affect the co-crystal configuration as well.

## Figures and Tables

**Figure 1 molecules-26-05536-f001:**
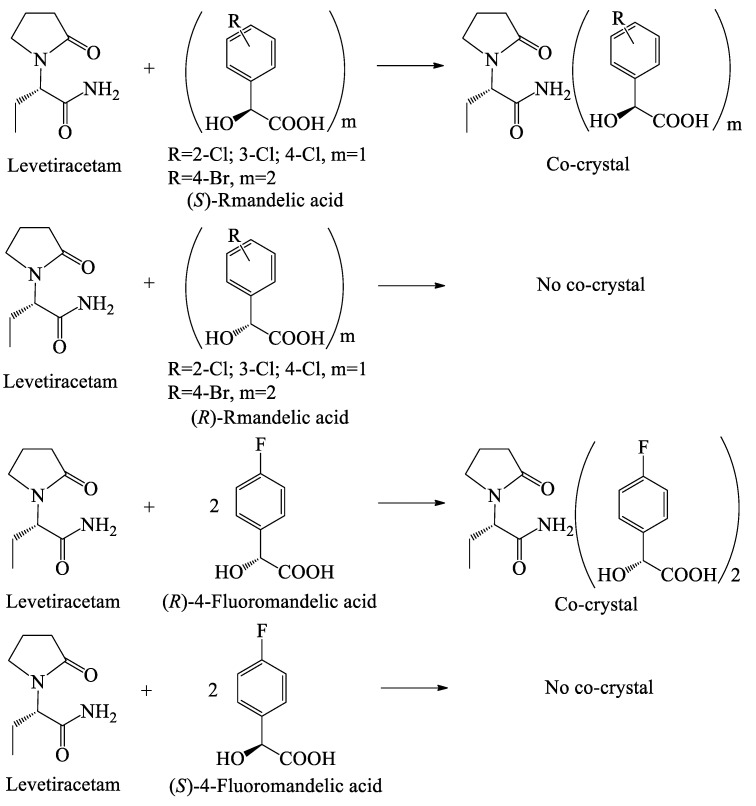
Enantiospecific co-crystal behavior between halogenated mandelic acid enantiomers and levetiracetam (LEV) [40].

**Figure 2 molecules-26-05536-f002:**
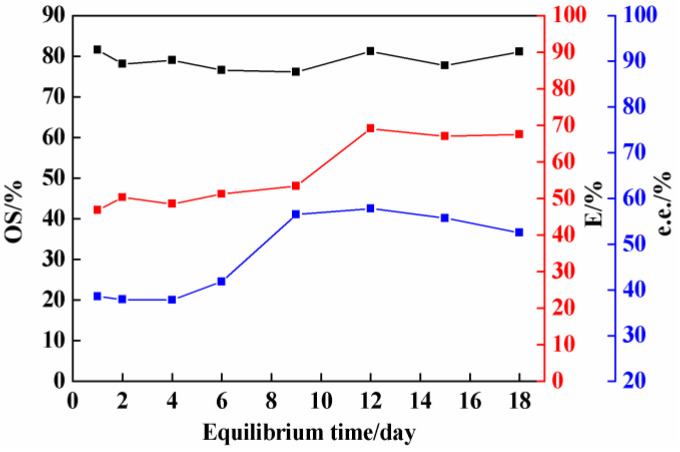
Effect of equilibrium time on the resolution of 3-chloromandelic acid (3-ClMA) by LEV (LEV = 2.58 mmol; 3-ClMA = 2.58 mmol; acetonitrile = 1.5 mL; crystallization temperature = −15 °C).

**Figure 3 molecules-26-05536-f003:**
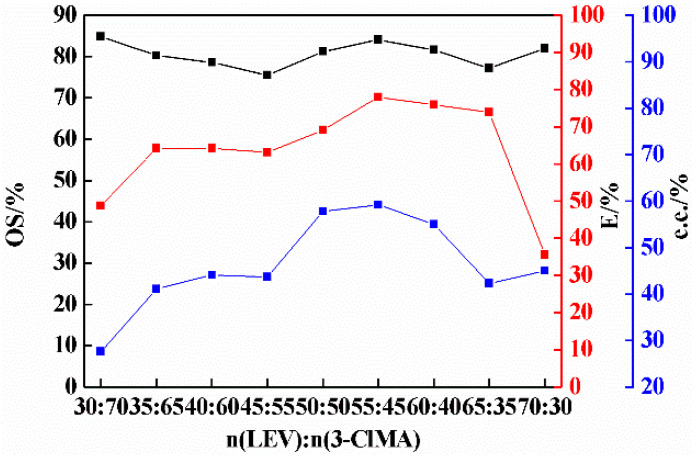
Effect of molar ratio on resolution of 3-ClMA by LEV (3-ClMA = 2.58 mmol; acetonitrile = 1.5 mL; crystallization temperature = −15 °C; equilibrium time = 12 days).

**Figure 4 molecules-26-05536-f004:**
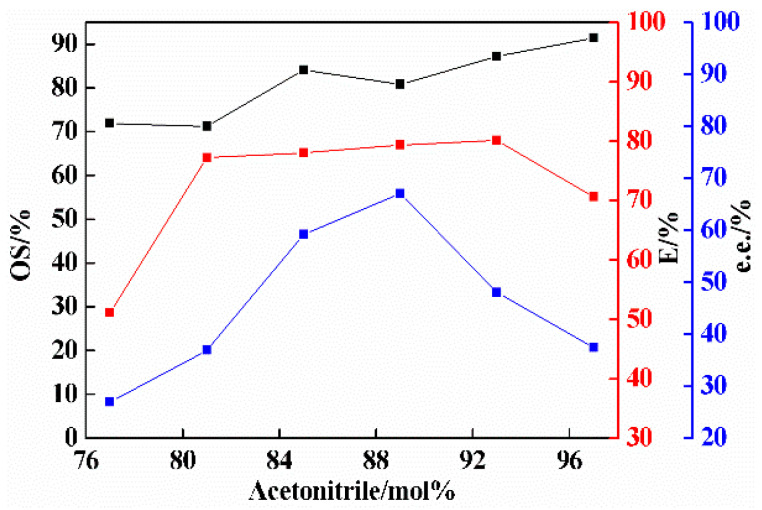
Effect of solvent amount on the resolution of 3-ClMA by LEV (the mole fraction of acetonitrile refers to the ratio of the moles of acetonitrile to the sum of the moles of acetonitrile and raw materials, mol%) [37,46]. (LEV = 2.84 mmol; 3-ClMA = 2.32 mmol; crystallization temperature = −15 °C; equilibrium time = 12 days).

**Figure 5 molecules-26-05536-f005:**
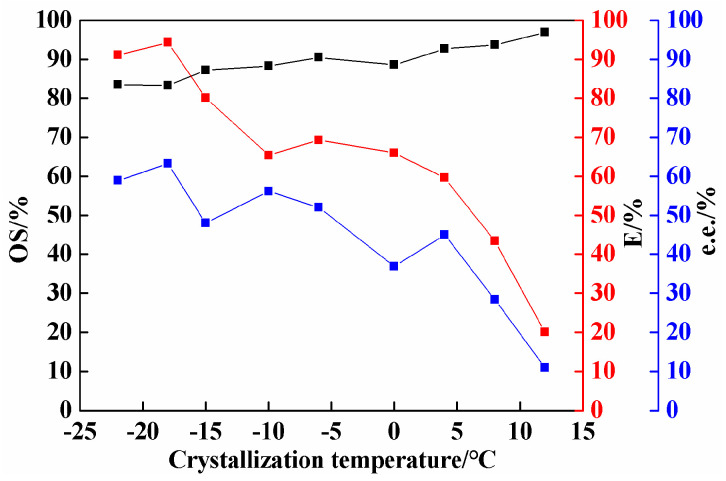
Effect of crystallization temperature on resolution of 3-ClMA by LEV (LEV = 2.84 mmol; 3-ClMA = 2.32 mmol; acetonitrile = 3.5 mL; equilibrium time = 12 days).

**Figure 6 molecules-26-05536-f006:**
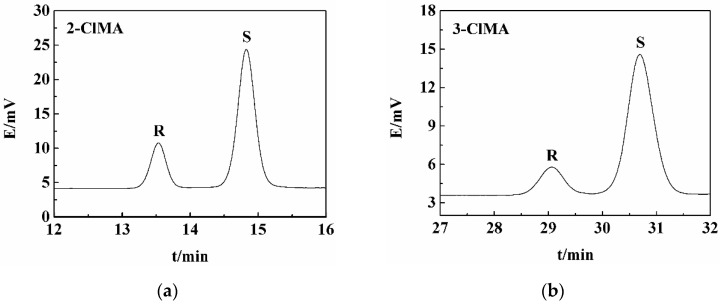
Typical liquid chromatograms of co-crystals obtained after resolving five halogenated mandelic acids with LEV: (**a**) 2-ClMA; (**b**) 3-ClMA; (**c**) 4-ClMA; (**d**) 4-BrMA; and (**e**) 4-FMA.

**Table 1 molecules-26-05536-t001:** Co-crystal resolution results of halogenated mandelic acids by LEV.

	Solid Phase (HPLC Analysis)	OS(OR)/%	E/%	e.e./% in Liquid Phase
2-Chloromandelic acid (2-ClMA)	LEV-(*S*)-2-ClMA	78	82	76
3-Chloromandelic acid (3-ClMA)	LEV-(*S*)-3-ClMA	83	94	63
4-Chloromandelic acid (4-ClMA)	LEV-(*S*)-4-ClMA	88	18	23
4-Bromomandelic acid (4-BrMA)	LEV-(*S*)-4-BrMA	71	28	21
4-Fluoromandelic acid (4-FMA)	LEV-(*R*)-4-FMA	90	54	29

The five processes of co-crystal resolution were completed under different resolution conditions. The reason is that the desired resolution effect cannot be obtained under the same conditions. The resolution conditions are described in Section 3.3, Section 3.4, Section 3.5, Section 3.6 and Section 3.7.

## Data Availability

The data presented in this study are available in Appendix A.

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
