# Peer review of "Resolution of Halogenated Mandelic Acids through Enantiospecific Co-Crystallization with Levetiracetam"

_molecules, 2021, doi:10.3390/molecules26185536_

Round 1

Reviewer 1 Report

The authors have attended satisfactorily all my observations.

Author Response

Point 1:The authors have attended satisfactorily all my observations.

Response 1: Thanks for the reviewer’ s comments. In addition, as reference 40 has not been officially published, we added “Supplementary materials” section to the manuscript, and presented the necessary data which is derived from unpublished reference 40 in the supplementary materials.

Reviewer 2 Report

As reference 40 has not been officially published, the authors should present the data missing in this present paper at least in the Supporting Information.

The data of Ref 40 are not, at the moment, available therefore the reader would not be available to access them.

Author Response

Point 1: As reference 40 has not been officially published, the authors should present the data missing in this present paper at least in the Supporting Information.

Response 1: Thanks for the reviewer’ s comments. According to your suggestion, we have added “Supplementary materials” section to the manuscript, and presented the necessary data which is derived from unpublished reference 40 in the supplementary materials.

Point 2: The data of Ref 40 are not, at the moment, available therefore the reader would not be available to access them.

Response 2: Thanks for the reviewer’ s comments. As reference 40 has not been officially published, we have uploaded reference 40, and we have translated Chinese into English for your convenience. And according to your suggestion, we put the required data in the supplementary materials of the manuscript, which is convenient for readers to read and check.

Round 2

Reviewer 2 Report

The authors added the missing material in the SI thus the paper can be accepted in the present form

This manuscript is a resubmission of an earlier submission. The following is a list of the peer review reports and author responses from that submission.

Round 1

Reviewer 1 Report

In this study, LEVETIRACETAM, a chiral pharmaceutical compound, is used as an optical resolution agent to optically resolve a target mandelic acid derivative by co-crystallization. In particular, the authors are investigating the optimal conditions for crystal preparation.
Generally, the optical resolution of pharmaceuticals is carried out using inexpensive optical resolution agents, which is the opposite to this study.  However, this study is interesting as an example of how optical resolution is achieved by co-crystal preparation.

Reference 40 is an important previous study, but it has not yet been published, so it is not an appropriate reference. This issue needs to be resolved.

It is important to note that co-crystals are not salts. It is necessary to clearly show that the combination of the two compounds in this study does not form a salt.

Also, if co-crystals were formed, their crystal structures should be clarified by PXRD or single crystal structure analysis. In general, co-crystals build a complicated system, including the ratio of components and the presence of polymorphs. Spectra alone would give some information about the co-crystals produced, but it is not enough. I believe that crystallographic analysis will allow for a more substantive discussion of optical resolution.

After optical resolution by co-crystal formation, the target molecules should be separated by a simple method and obtained pure. The authors should mention this method in the conclusion section. 

Also, the authors should clearly conclude the significance of using levetiracetam as an optical resolution agent.
Is the optical resolution ability of this optical resolution agent higher than other compounds? i.e., whether it is a suitable compound for optical resolution?

Reviewer 2 Report

The paper from Wang and Peng described the resolution of substituted mandelic acids by co-crystallization with the antiepiletic drug Levetiracetam.

The data presented arised some concerns in the reviewer:

-the scope is quite limited; only 5 examples are presented. The scope should be enlarged including mandelic acid with other sunstituents.

-The authors seeded the solution with a co-crystal of Levetiracetam and chiral mandelic acid. Have the authors tried to seed the solution with the only chiral mandelic acid? This should be an important control to run to assess the need for Levetiracetam

-The experimental part is really poor. The authors should improve it, like presentig the HPLC traces of racemic mixtures, including the characterization of the co-crystal, etc

-Have the authors studied the co-crystal stucture? Which kind of interactions are present between mandelic acid and  Levetiracetam?

-How the authors can justify the inverted selectivity for 4-FMA? The structure of the co-crystal should be studied in detail.

For the reasons above, the paper should not be accepted in the present form.

Reviewer 3 Report

This manuscript reports on the results achieved from racemic resolution experiments of diverse halogen-containing mandelic acids (MA) using a strategy based on the formation of cocrystals with enantiopure levetiracetam (LEV). The authors have shown previously (see ref 40, which is apparently not published yet) that LEV forms enantiospecific cocrystals with either the S- or R-enantiomer of the mandelic acids studied. In the present contribution, mainly different experimental conditions and parameters were evaluated in depth to establish ideal conditions for the racemic resolution. By optimizing these parameters in one case (with 3-ClMA), high resolution efficiency and enantiomeric excess could be achieved.

I enjoyed reading the manuscript which is well-presented and recommend acceptance if the following issues are attended:

  1. Introduction, lines 47-52. Aside from the cited option for resolution through cocrystallization with an enantiopure reagent (= only one enantiomer from the racemate forms a cocrystal) there is the second option that both enantiomers form each a cocrystal. There is even the third option that the cocrystal contains both enantiomers, which would not give resolution. This should be mentioned and properly cited (see e.g. ref 37-38 and references cited therein).
  2. Introduction, line 68. The reader would probably like to be informed about the synthesis, availability and costs of levetiracetam.
  3. Lines 77 and 81: delete “enantiospecifically” since for the reactions mentioned only one enantiomer was used instead of the racemate.
  4. Figure 1 and related text: It should be mentioned how the results of the reactions were evidenced. PXRD analysis?
  5. Results and Discussion, lines 95-105. Please mention briefly the procedure used for the resolution and methods employed for the determination of the optical purity of the target enantiomer.

Further, there is an important aspect that needs to be attended in the context of observation No.  1: Upon combining LEV with the target compound of the resolution process, starting from the racemic form instead of the enantiopure forms can lead to different results in the cocrystallization processes. This applies particularly in the case when the crystal structure and, thus, the crystal lattice energy of the racemate (RS-form) is different from that of the compound in enantiopure form. If the racemate of the mandelic acids used herein are not just physical mixtures of crystals of the R- and S-forms, but are instead both embedded in single crystal lattices, it is necessary to realize the reactions outlined in Figure 1 also with the racemate (not only with the individual enantiomers). PXRD analysis of the cocrystallization product(s) will show if the reaction proceeded in the same form as for the individual enantiomers or resulted in different products when starting from the racemate. For background, see references 37-38 and also Crystal Growth Des. 18, 7356 (2018), and literature cited therein.

  1. Section 2.3. The type of solvent should also influence the process. A comment in this direction should be included and some explanation why MeCN was chosen finally.
  2. Section 2.4. The expression “cooling temperature” caused some confusion to me. “Crystallization temperature” or “temperature for crystal growth” might be more adequate.

Further, at this point I suggest explaining briefly which temperature was used for the cocrystallization process before storage and cooling.

  1. Table 1: I suggest to incorporate also the results for 3-ClMA. Further, the information in the note added at the end of the table is not clear to me.
  2. Section 3: I suggest moving this section to the end of the manuscript or before the Results and Discussion section.
  3. Section 3.2., lines 200-209: please indicate ranges for the quantities used of each component. Also include the temperature range explored and the time used for the crystallization process.
  4. Figure 6. Please be more precise in the legend regarding the origin of the chromatograms. Please indicate also which peak corresponds to which enantiomer.
  5. Equation 4: some parentheses seem to be missing.
  6. Lines 276-281: Please be more precise explaining what is meant precisely with “separation” in each case.
  7. Conclusions: Probably I got confused, but I thought that 3-ClMA gave the best results (not 2ClMA). Please revise.

Line 290: the number for the %ee is missing.

  1. References 18 and 38. Please revise the citing of the names of the authors: The first names are spelled fully instead of the family names.